# RNA covariation at helix-level resolution for the identification of evolutionarily conserved RNA structure

**Elena Rivas** *

Department of Molecular and Cellular Biology, Harvard University, Cambridge, Massachusetts, United States of America

* elenarivas@fas.harvard.edu

## Abstract

Many biologically important RNAs fold into specific 3D structures conserved through evolution. Knowing when an RNA sequence includes a conserved RNA structure that could lead to new biology is not trivial and depends on clues left behind by conservation in the form of covariation and variation. For that purpose, the R-scape statistical test was created to identify from alignments of RNA sequences, the base pairs that significantly covary above phylogenetic expectation. R-scape treats base pairs as independent units. However, RNA base pairs do not occur in isolation. The Watson-Crick (WC) base pairs stack together forming helices that constitute the scaffold that facilitates the formation of the non-WC base pairs, and ultimately the complete 3D structure. The helix-forming WC base pairs carry most of the covariation signal in an RNA structure. Here, I introduce a new measure of statistically significant covariation at helix-level by aggregation of the covariation significance and covariation power calculated at base-pair-level resolution. Performance benchmarks show that helix-level aggregated covariation increases sensitivity in the detection of evolutionarily conserved RNA structure without sacrificing specificity. This additional helix-level sensitivity reveals an artifact that results from using covariation to build an alignment for a hypothetical structure and then testing the alignment for whether its covariation significantly supports the structure. Helix-level reanalysis of the evolutionary evidence for a selection of long non-coding RNAs (lncRNAs) reinforces the evidence against these lncRNAs having a conserved secondary structure.

**Data Availability Statement:** The aggregated p-value method has been implemented in the software program R-scape version 2.0.0.p (December, 2022). Using any of the options to

## Author summary

In this manuscript, I introduce a new measure of significant structured RNA covariation at helix level. I show that helix-level covariation is more sensitive and robust than that obtained at base pair level in order to identify evolutionarily conserved RNA structure. Helix-level covariation exposes the circularity of using proposed RNA structures to build alignments in which to test for structural covariation. Helix-level covariation reinforces the evidence against some lncRNAs with proposed structures having a conserved secondary structure.

report the helix-aggregation methods, R-scape produces a file.helixcov with all the information about the helix-level analysis. Details are given in the R-scape user guide provided in the supplemental material. The code is available from the supplemental materials of this manuscript, from the R-scape website eddylab.org/R-scape/, and from the lab website rivaslab.org. Data used in this manuscript, including alignments, are part of the supplemental material (referred to as S1 Data in the manuscript) http://rivaslab.org/publications/Rivas23a/supplemental_material.tar.gz.

**Funding:** This work has been partially supported by NIH grant R01-GM144423 (to E.R.). The funders had no role in study design, data collection and analysis, decision to publish, or preparation of the manuscript.

**Competing interests:** The author has declared that no competing interests exist.

## Introduction

The identification of conserved RNA structure is an important tool to discover RNA transcripts performing biologically relevant and novel functions. Conserved RNA structure results in covariation observed between base paired positions in alignments of homologous sequences. Because a conserved RNA structure is not the only source of covariation, a statistical test is required to distinguish covariation due to RNA structure from covariation due to other sources. Because the space of RNA transcripts without an assigned function is large, it is important to build robust methods that provide stringent control of false positive rates.

To that effect, R-scape [1, 2] implements a null hypothesis of covariation due to homology not associated with RNA structure. For each pair of columns in an alignment, R-scape assigns a p-value that reflects the probability of having obtained that covariation score or higher under the null hypothesis of covariation due solely to the phylogenetic relationships between the sequences. The goal of this work is to improve the sensitivity in the detection of conserved RNA structures while maintaining the specificity achieved by the R-scape statistical test of significant base-pair covariation.

R-scape calculates covariation between all pairs of position in an alignment, and p-values are obtained by treating pairs as independent from each other. However, RNA base pairs, especially Watson-Crick base pairs (which typically show more covariation) do not occur in isolation, but they stack on each other forming helices of variable length, typically ranging from 3 to 25 base pairs with a median of 12 in Rfam 14.9 consensus structures (for an operational definition of helix that allows bulges and internal loops with at most two residues). As it has been observed before [3], I hypothesize that a statistical test that assigns significance at helix level should be more biologically relevant towards the identification of conserved RNA structure. How to transition from base-pair p-values into more powerful and yet robust helix-based p-values is the subject of this work.

Interestingly, essentially the same problem arises in RNA-seq differential expression analysis, where expression data are obtained for individual RNA transcripts, but it is more robust and experimentally relevant to perform the differential expression analysis at gene level using aggregation of transcript p-values [4]. Here following a similar approach, I present a novel way to study RNA significant covariation at helix level by aggregating the original p-values obtained at the base-pair level. I test different aggregation methods: Fisher [5], Lancaster [6], Šidák [7], and a weighted-Fisher method [8]. The Lancaster and weighted-Fisher methods, in addition to covariation p-values (sensitivity), also use covariation power (specificity) to weight the base-pair p-values.

This work shows that the analysis of significance performed by R-scape at the base pair level, followed by a helix-level meta analysis improves sensitivity in detecting evolutionary signals of conserved RNA structure without compromising the stringent specificity achieved by simple base pair significance. Using helix-level covariation and two long non-coding RNAs (lncRNAs) with proposed RNA structures, COOLAIR [9, 10] and NEAT1 [11, 12], I also show the risks of assessing significant covariation using alignments that have been built using covariation in the first place. This effect is important when evaluating the potential of a conserved RNA structure in uncharacterized heterogeneous transcripts.

## Results

### Significant covariation: From base pair to helix

I calculate aggregated p-values in order to obtain covariation inference at helix level from the significant covariation observed at base pairs level. I have evaluated three classical aggregation

methods: Fisher's [5], Lancaster's [6], Šidák [7], amongst many more existing [13], and a newer weighted Fisher method [8]. Details of the different aggregations methods are given in the Methods.

The Fisher method [5] considers all p-values to aggregate $(p_1, \ldots, p_N)$ as equivalent. Under the null hypothesis of independent and identically distributed p-values, the statistic $T_f = \sum_{n=1}^{N} -2 \log(p_n)$ is chi-squared distributed with $2N$ degrees of freedom.

The Lancaster method [6] generalizes Fisher's by weighting the p-values relative to each other. The Lancaster method assigns a positive integer weight $w_n$ to each p-value $p_n$, and uses the statistic $T_l = \sum_n F_{\chi^2_{w_n}}^{-1}(p_n)$, where $F_{\chi^2_{w_n}}^{-1}$ is the inverse cumulative density function (CDF) of a chi-square distribution with $w_n$ degrees of freedom. For independent and identically distributed p-values, $T_l$ is chi-squared distributed with $\sum_n w_n$ degrees of freedom.

The Lancaster weights allow one to incorporate additional meaningful biological information into the analysis. For instance, in differential gene expression analysis, the weighting schema is based on transcript counts [4]. In our case to aggregate alignment-derived R-scape base-pair p-values, each base-pair p-value is weighted by the number of substitutions observed for those two positions in the alignment. This variational information is complementary to the pairwise covariation measured by the p-values. Covariation requires variation, but the absence of covariation can occur either associated with or in the absence of variation, leading to two different scenarios. While lack of variation renders the analysis inconclusive as to the presence of a conserved RNA structure or not, variation in the absence of covariation is evidence against a conserved structure [2].

I have also tested a generalization of the Lancaster method that allows the weights to be arbitrary real numbers instead of just integers, that I name the weighted-Fisher method. The weighted-Fisher method relies on the chi-square distribution being a particular case of the Gamma distribution. The weighted-Fisher method uses the statistic $\sum_n F_{\Gamma(w_n/2, 1/2)}^{-1}(p_n)$ which under the null hypothesis follows a Gamma distribution $\Gamma(\sum_n w_n/2, 1/2)$. Details are provided in the Methods. For the weighted-Fisher method, I used the base pair covariation power as weights. Covariation power, reported by the R-scape method, measures the probability that a base pair would be detected as significantly covarying based on the number of observed substitutions. Covariation power is a normalized version of the substitution weights used by the Lancaster method.

I have also implemented the Šidák method [7] that uses a test based on the minimum p-value (Methods).

For all methods, R-scape reports an aggregated helix E-value correcting for multiple testing that is obtained by multiplying the corresponding helix p-value by the number of helices in the proposed structure. The aggregated helix E-value reports the expected number of helices with that E-value or higher under the null hypothesis of covariation not due to RNA structure.

*P-value aggregation usually results in increased sensitivity.* A collection of base pairs all with borderline covariation evidence may result in a more robust helix signal as result of the aggregation of all the individual marginally significant p-values (Fig 1a). For the Rfam dataset [14], 6% (144/2,374) of all helices become significant in the absence of any individual significant base pair using the Lancaster method. That fraction lowers to 5.6%, 5.2% and 0.7% for the weighted-Fisher, Fisher and Šidák methods respectively.

Moreover, aggregation does not reduce sensitivity as one single base pair with strong covariation signal usually renders the whole helix significant even if the rest of the base pairs in the helix have non significant p-values (example given in Fig 1b). Although that is not always the case as reported in the example in Fig 1c, but those are not common occurrences. For the

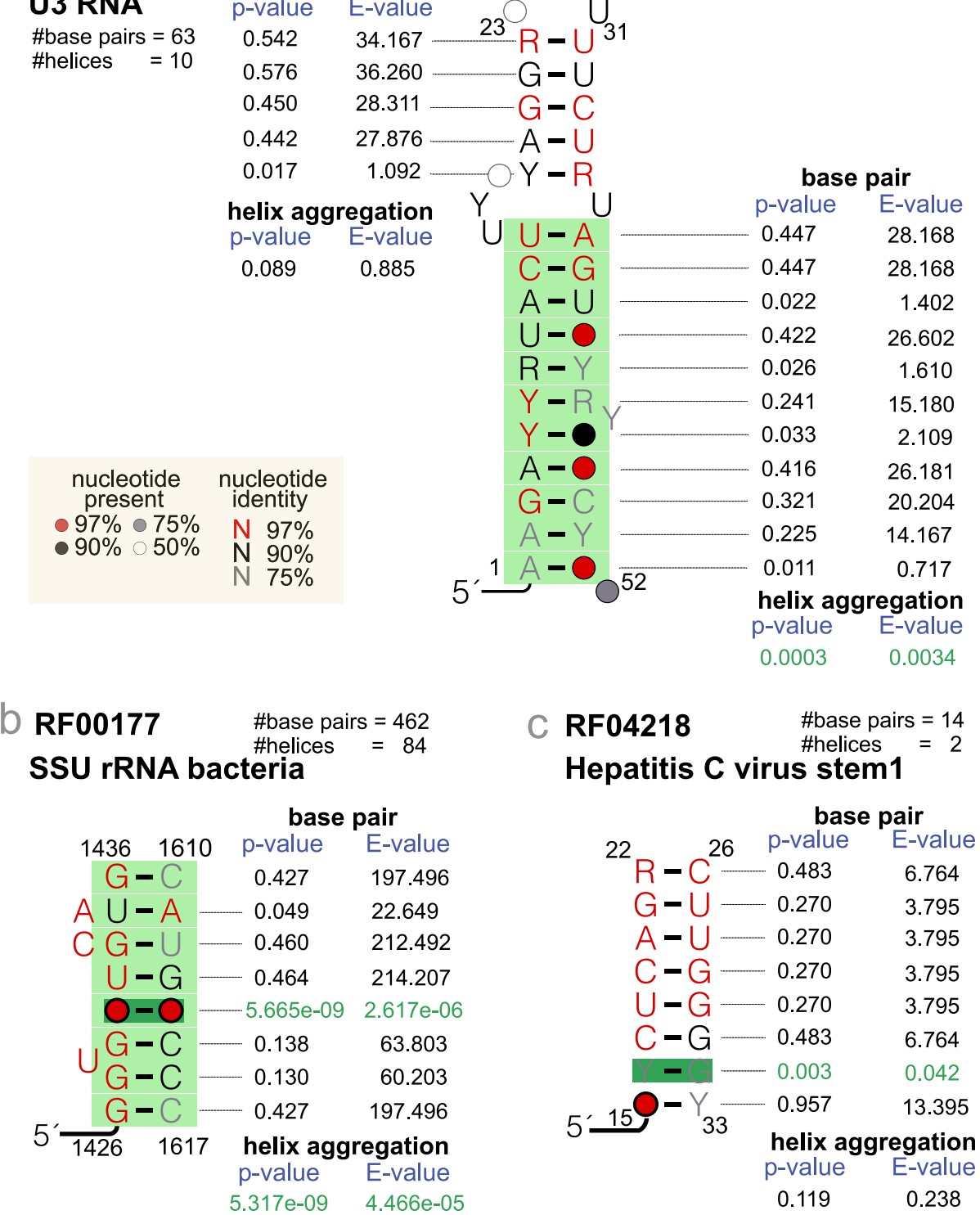

**Fig 1. Examples of base-pair versus helix-level tests of significant covariation. (a)** Example of two helices from the same structure both without significantly covarying base pairs, one resulting in a non-significant helix (top), the other resulting in a significant helix (bottom). **(b)** Example where one significant base pair alone (base pair E-value < 0.05) often results in a significant helix with a significant E-value (helix aggregated E-value < 0.05). **(c)** Example of a helix with one significant base pair that does not significantly covary. For the Rfam dataset described in Methods, only 1% (4/363) of the helices with one significant base pair result in a helix that does not significantly covary. Base pair

p-values are converted to base pair E-values after correcting by the number of base pairs tested in the alignment. Helix aggregated p-values are converted to helix E-values after correcting by the number of helices in the proposed structure. Aggregated helix p-values are calculated with the Lancaster method [6]. Significantly covarying base pairs and helices are depicted in dark and light green respectively. All examples have been obtained from the Rfam seed alignments. Helix coordinates correspond to the column positions in the corresponding Rfam alignment.

Rfam-derived structural dataset described in Methods, only 1% (4/363) of the helices have one or more significantly covarying base pair but result in a non significant helix.

However, this reported increased helix p-value sensitivity could occur at the expense of lower specificity. We need the method to also be robust by maintaining low false positive rates. Next, I introduce a benchmark that allows us to jointly test the sensitivity and specificity of the different aggregation methods.

### A sensitivity/specificity benchmark

An aggregation scheme could compromise the specificity of the R-scape method. Specificity is very important in the identification of transcripts with conserved RNA structure. Especially given the large size of the vertebrate transcriptome, I want to keep the number of false positives as low as possible without sacrificing sensitivity. I have created a benchmark to assess the sensitivity/specificity of the helix-level significant covariation test.

I have created a benchmark of **real structural RNAs**, and another one of **synthetic structural RNAs**. Each benchmark includes a set of **positive** real/synthetic structural RNA alignments, as well as a set of **decoy** RNA alignments that lack structure.

The real benchmark includes as trues a subset of 326 Rfam structural alignments and consensus structures, selected by their accuracy according to specifications provided in the Methods. The true synthetic structural alignments are simulated by following the phylogeny and structure of each of the 326 RNA families using the program R-scape-sim [1]. True synthetic alignments were created at three different evolutionary distances.

I use two kind of decoy alignments. For the synthetic benchmark, I generated decoy alignments that start with an inferred root sequence/structure that is then evolved with the same phylogeny as the true alignments, but constructed without any structural constraints. For the real benchmark, I also added another set of decoys which are the null alignments internally generated directly from the input alignment by the program R-scape to test the hypothesis of covariation due to phylogeny alone. Both sets of decoy alignments remove any structural constraints but are otherwise similar to the input alignment (details comparing the alignments are provided in Table 1 and S1 Fig).

Each Rfam seed alignment includes a consensus RNA structure which is a collection of annotated base pairs common to all sequences in the alignment. Starting from one seed Rfam alignment, all four type of alignments in the benchmark share the same consensus structure.

**Table 1. The real and synthetic structured RNA benchmarks.** The real positive alignments are Rfam seed alignments. The synthetic alignments were generated using the program R-scape-sim from the Rfam positives; synthetic positives follow the phylogeny and preserve the secondary structure of the RNA family; synthetic decoys only preserve the phylogeny. The R-scape null decoys are the null alignments generated by R-scape and saved using the option --outnull. Details of all the alignments (such as length and average percentage identity) and their associated structures (number of base pairs, helices and helix length) are given in S1 Fig.

| benchmark | positives | | | | decoys | | | |
|---|---|---|---|---|---|---|---|---|
| | *alignments* | *helices* | *base pairs* | *provenance* | *alignments* | *helices* | *base pairs* | *provenance* |
| synthetic RNAs | 19,079 | 93,892 | 562,112 | synthetic structural | 19,068 | 93,849 | 561,097 | synthetic phylogeny |
| real RNAs | 326 | 2,374 | 13,915 | Rfam | 19,068 | 93,849 | 561,097 | synthetic phylogeny |
| | | | | | 12,680 | 57,500 | 336,140 | R-scape null |
| | | | | | 31,748 | 151,349 | 897,237 | totals |

While the decoy alignments are not constructed using the structural constraints of the true Rfam alignment, the RNA consensus structure is just directly assigned to the alignment extant sequences to produce decoy structures.

The annotated base pairs in the positive alignments constitute the set of true base pairs. The remaining possible base pairs in these positive alignments are ignored because they could be unannotated true base pairs. All annotated pairs in the given consensus structure of the decoy alignments constitute the set of decoy base pairs. Helices are extracted from the true and decoy structures by splitting contiguous sets of stacked base pairs separated by more than two unpaired nucleotides. For instance,

$$\begin{matrix} A & A & G & G & A & G & G & A & G & G & A & A \\ & & | & | & & | & | & & | & | & & \\ A & A & C & C & A & C & C & A & C & C & A & A \end{matrix}$$

contains one helix with six base pairs.

Additional details and characteristics of the real and synthetic benchmark true and decoy alignments are described in the Methods and in S1 Fig.

**Helix covariation increases sensitivity/specificity in structured RNAs.** For a given benchmark, all alignments (positives and decoys) are tested using R-scape. Each base pair gets associated a p-value, and the p-values of the base pairs constituting a helix are used to calculate an aggregated helix p-value for each annotated helix. Helix aggregated p-values are converted to E-values by correcting for the multiple helix testing in the alignment. A helix E-value estimates the expected number of false helices that would have such E-value or larger. Results of the R-scape covariation analysis for all alignment helices are merged and sorted by helix E-value. The ranked list of helices is used to calculate a plot of fraction of true positive helices detected at increasing thresholds of mean false positives per query alignments, ranging from 0.0001 to 1.

Fig 2 shows sensitivity/specificity results for the synthetic and real RNAs benchmarks. I present results for the four aggregation methods tested: Fisher, Lancaster, weighted-Fisher and Šidák. In both benchmarks, I observe that the Lancaster method performs best, although the difference from the Fisher and weighted-Fisher methods is not large.

As a baseline, the benchmark compares the results obtained using aggregated helix E-values to an unaggregated method. The $k$-unaggregated method considers a helix supported if it has at least $k$ significantly covarying base pairs (base-pair E-value < 0.05). I observe that all methods except for Šidák report higher sensitivity than a given $k$-unaggregated method for the same number of false positives.

For the 1-unaggregated method, 78.4%/92.7% of the true helices (for the real/synthetic benchmarks respectively) are reported to have at least one significant base pair for an estimated 0.0465/0.0231 observed false positive helices per alignment. In contrast, the Lancaster method achieves the same sensitivity for a much smaller expected number of false positives estimated at 0.0007/0.0001 in the real/synthetic benchmarks. Equivalently, the Lancaster method achieves the same specificity as the unaggregated method with an increase in sensitivity from 78.4% to 84.0% in the real benchmark, and from 92.7% to 97.9% in the synthetic benchmark.

The base-pair p-values used for aggregation in Fig 2a and 2b are calculated using R-scape's two-set test in which two different tests are performed (and two different sets of p-values are collected) depending on whether the base pair is in the proposed structure or not. For completeness, I have also performed a similar benchmark in which the R-scape p-values are calculated using the one-set test in which the annotated structure is ignored and all pairs are tested equally (S2 Fig). As expected, I observe that sensitivity lowers, but the rest of conclusion of the benchmark remain unchanged: aggregation gives better sensitivity to specificity ratio than not performing any aggregation, and the Lancaster method slightly outperforms all other tested methods.

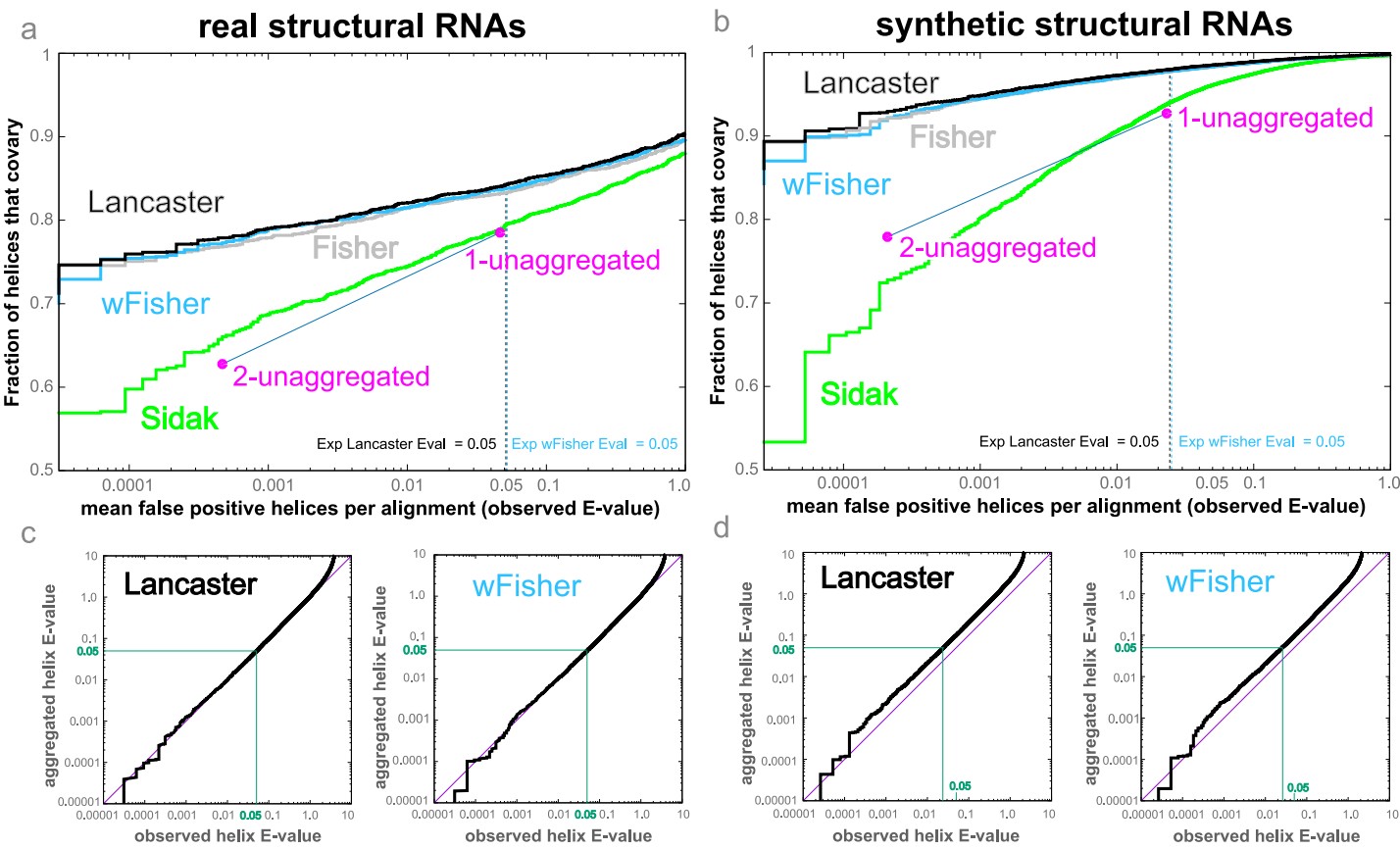

**Fig 2. Benchmark of helix-level covariation sensitivity and specificity. (a, b)** Sensitivity/specificity plots for the real and synthetic benchmarks respectively of the four aggregation methods plus two k-unaggregated methods ($k = 1, 2$). The y-axis is the fraction of true positive helices detected with an aggregated E-value better than the number of false positive helices per query alignment specified on the x-axis (observed helix E-values). The dotted lines indicates the mean number of false positive helix per alignment with an estimated helix E-value $\leq 0.05$. For the real benchmark the expected E-value of 0.05 intersects with the x-axis (observed helix E-value) at almost exactly the same value. For the synthetic benchmark the expected E-value of 0.05 occurs at a lower observed E-value of 0.022. **(c, d)** Helix E-value accuracy plots. The observed helix E-values are plotted against the calculated aggregated E-values for the Lancaster and weighted-Fisher methods.

Similar benchmark results are obtained when I measure specificity using the false discovery rate (FDR) which calculates the fraction of helices with E-values smaller than a given E-value threshold that are decoys (S3 Fig).

**Accuracy of helix E-value determination.** These benchmarks also provide an empirical test of how accurately the estimated aggregated helix E-values reproduce the observed E-values. Fig 2c and 2d show a comparison of the aggregated helix E-values to the observed mean false positive per alignment based on the decoys. We observe an excellent correspondence for the real RNAs benchmark (Fig 2c). For the synthetic RNAs benchmark (Fig 2d), the small disagreement goes in the conservative direction as the observed aggregated E-values are lower than the estimated ones.

**Helix-level analysis is more robust than base-pair-level significance.** The benchmark also allows us to calculate sensitivity/specificity plots for the covariation analysis at base-pair level, where results in all alignments (positive and decoy) are merged and sorted by base-pair E-value. Results are presented in S4 Fig. I observe that helix aggregation outperforms base-pair level covariation analysis as it provides greater sensitivity at lower false positive rates. For instance, for the synthetic benchmark, 99.0% helix detection occurs for 0.12 false positive

helices per alignment on average (Fig 2a), which corresponds to approximately 1.4 false positive base pairs (for a median of 12 base pairs per helix reported in S1 Fig). On the other hand, it takes on average 12.2 false positive base pairs to detect 99.0% of the base pairs without aggregation (S4(a) Fig).

We also observe that the base-pair E-values also reproduce accurately the observed mean number of base-pair false positive base pairs per alignment (S4(c) and S4(d) Fig).

## Structural evidence on lncRNAs based on helix covariation

The investigation of possible functions for lncRNAs ordinarily leads to the question of whether they include conserved RNA structures. Structures have been proposed along with alignments said to provide evolutionary support in lncRNAs such as: Cyrano RNA [19], HOTAIR [20], MEG3 [21], NEAT1 [12], XIST [22–24], lincRNAp21 [25], ncSRA [26] in vertebrates, and COOLAIR in plants [10, 27] amongst others. Here, I examine the evidence provided by those alignments using helix-level covariation analysis.

Fig 3 compares results for those lncRNAs to the structural RNAs used in the benchmark. Fig 3a reports the number of base pairs that covary significantly as a function of the number of base pairs expected to covary given the variation observed in the alignment (covariation power) [1, 2]. Fig 3b depicts helix-level covariation also as a function of covariation power.

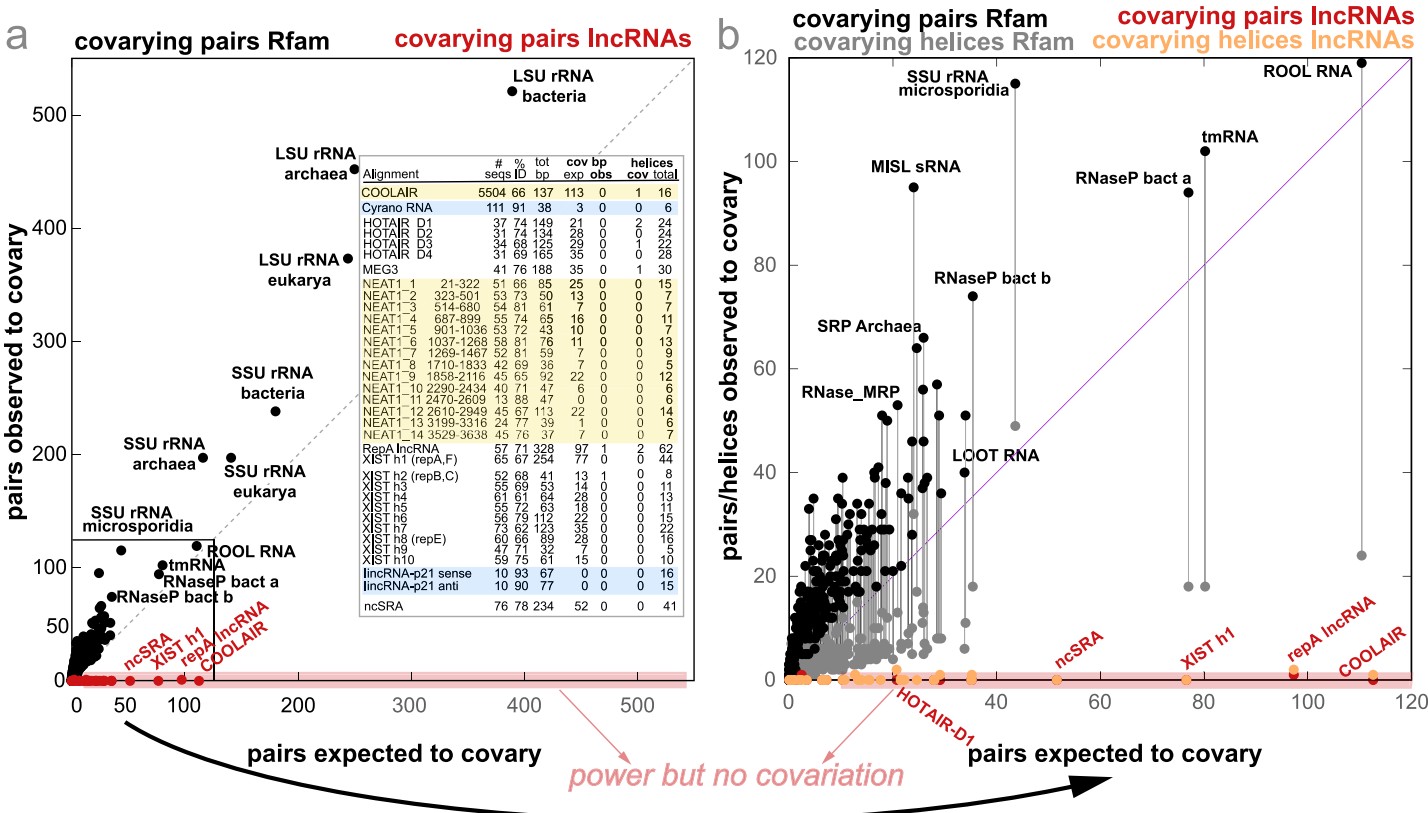

**Fig 3. Covariation evidence for 362 structural RNAs and 8 lncRNAs.** (a) Base-pair covariation versus power of covariation (black for structural RNAs, red for lncRNAs). (b) Helix aggregated covariation using the Lancaster method versus power of covariation (gray for structural RNAs, orange for lncRNAs). **Insert:** highlighted in yellow, lncRNAs COOLAIR and NEAT1 showing results for nhmmer alignments; in blue, Cyrano RNA and lincRNAp21 without enough power to make any evolutionary inference. Not highlighted are the remaining four lncRNAs HOTAIR, MEG3, XIST and ncSRA with extensive power but minimal covariation. All structural alignments are provided as part of S1 Data. Details of the alignment's provenance are given in the Methods.

I summarize the results as follows,

*COOLAIR shows evidence of protein-coding evolution but not of an RNA structure.* COOL-AIR is an *Arabidopsis* non-coding transcript originating from the 3′ end of the gene encoding the floral repressor flowering locus C (FLC) [9]. A predicted COOLAIR structure [10, 27] was originally supported by a comparison of only six plant species [10], and resulted in an alignment with low power that reported one significantly covarying base pair (E-value: 0.048) [2]. The helix analysis of this low-power alignment reports two covarying helices without any significant base pair (Lancaster E-values: 0.0054 and 0.022). This result, combined with the fact that the expected number of covarying pair by covariation power matches the number of observed covarying pairs encouraged us to build deeper COOLAIR alignments.

I created an nhmmer [28] alignment for the COOLAIR class II.i, one of the COOLAIR isoforms for which structures have been proposed [27], searching for homologs of the *Arabidopsis thaliana* sequence in other plant genomes. The resulting nhmmer alignment does not support the proposed COOLAIR class II.i structure (Fig 3 Insert). On the other hand, the nhmmer alignment reports 12 pairs that significantly covary between positions that are either contiguous or separated by just a few residues. These significantly covarying pairs that do not form any RNA helix are the result of another signal present in the alignment which results from the fact that COOLAIR class II.i isoform is antisense to the first exon of FLC. Variability within codon positions in alignments of protein-coding gene exons results in significant covariation above phylogenetic expectation that I have characterized elsewhere [29].

*NEAT1 lncRNAs lack evidence of a conserved structure.* The NEAT1 lncRNA are involved the formation of paraspeckles nuclear bodies [11]. For the short-isoform NEAT1_S lncRNA, structures have been proposed and alignments have been produced after dividing the sequence into 13 segments [12]. An analysis of these alignments shows different levels of evidence both at the base-pair and helix level. Fig 4a shows the analysis for the structural alignment of one of the NEAT1 RNAs (region 323–502) that shows stronger covariation. However, I have evidence to argue that the covariation in Fig 4a is spurious and the result of forcing a structural alignment.

I show the point in Fig 4b, where I present a different structural alignment of the same NEAT1_323–502 sequences, but this time aligned according to a different structure with similar number of base pairs but where the base pairs are all different from those in the published structure. Both disjoint structural alignments have similarly large amount of covariation (Fig 4a and 4b). Because these are two disjoint structures–and one of them completely arbitrary, it follows that the covariation observed is not conclusive of any conserved structure, but likely to be caused simply by aligning to a covariance model (profile SCFG) that enforces a particular structure. To further support this point, alternative non-structural alignments created using the methods nhmmer [17] and MUSCLE [18] result in alignments that do not report any covariation for the proposed base pairs (Fig 4c).

An increasingly popular way of creating alignments to assess for the presence or not of a conserved RNA structure is actually assuming a structure, and creating a structural alignment based on it [30–32]. RNA structural alignments, in particular those created with the profile stochastic context free grammars used by Infernal [16] have proven excellent tools to classify *known* structural RNAs, and they are the foundation of structural databases such as Rfam [14]. However, using a *not confirmed* structure to produce an alignment to confirm the structure introduces a circularity (or double dipping [33, 34]) that should be avoided.

This effect seems to be even more important for helix-level covariation as a small effect per base pair adds up over a whole helix. As another example, I have created two structural alignments for COOLAIR, one using the published structure [27], the other using an alternative structure with comparable number of base pairs but all completely different from the

## NEAT1 323-501

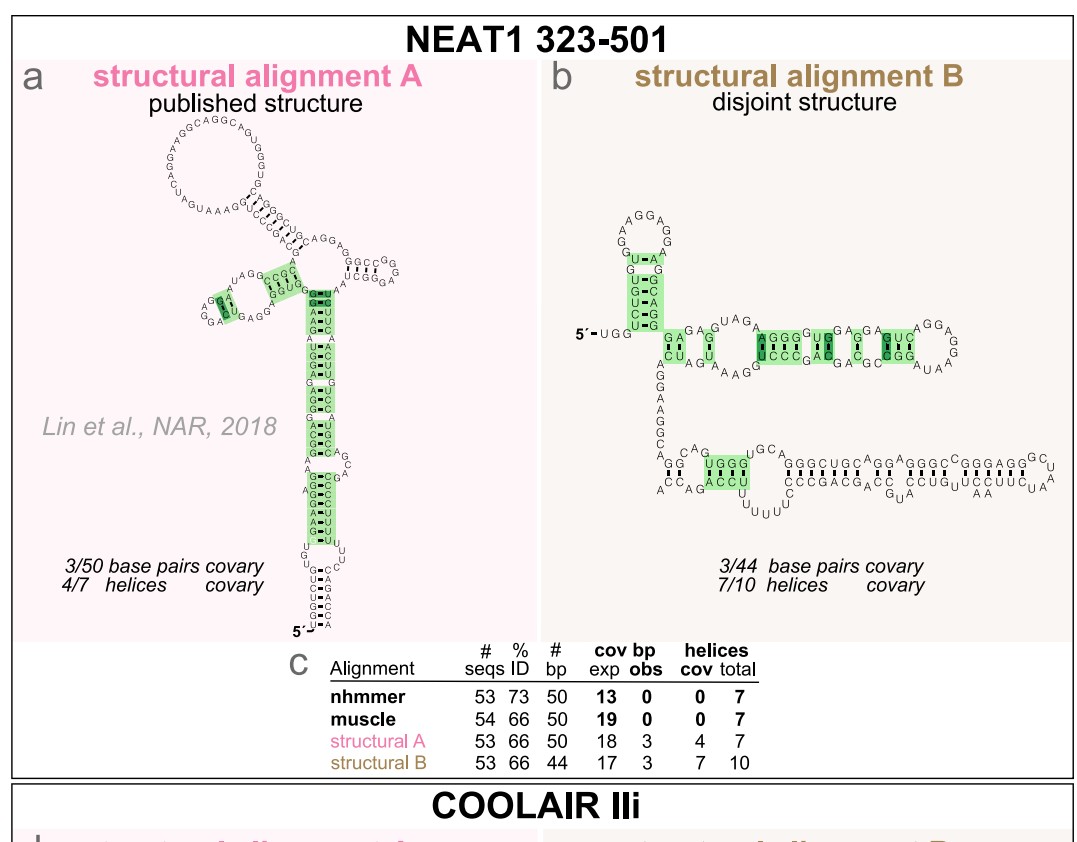

**a** structural alignment A
published structure

*Lin et al., NAR, 2018*

3/50 base pairs covary
4/7 helices covary

**b** structural alignment B
disjoint structure

3/44 base pairs covary
7/10 helices covary

**c**

| Alignment | # seqs | % ID | # bp | cov bp exp | cov bp obs | helices cov | helices total |
|---|---|---|---|---|---|---|---|
| **nhmmer** | 53 | 73 | 50 | **13** | **0** | 0 | 7 |
| **muscle** | 54 | 66 | 50 | **19** | **0** | 0 | 7 |
| structural A | 53 | 66 | 50 | 18 | 3 | 4 | 7 |
| structural B | 53 | 66 | 44 | 17 | 3 | 7 | 10 |

## COOLAIR IIi

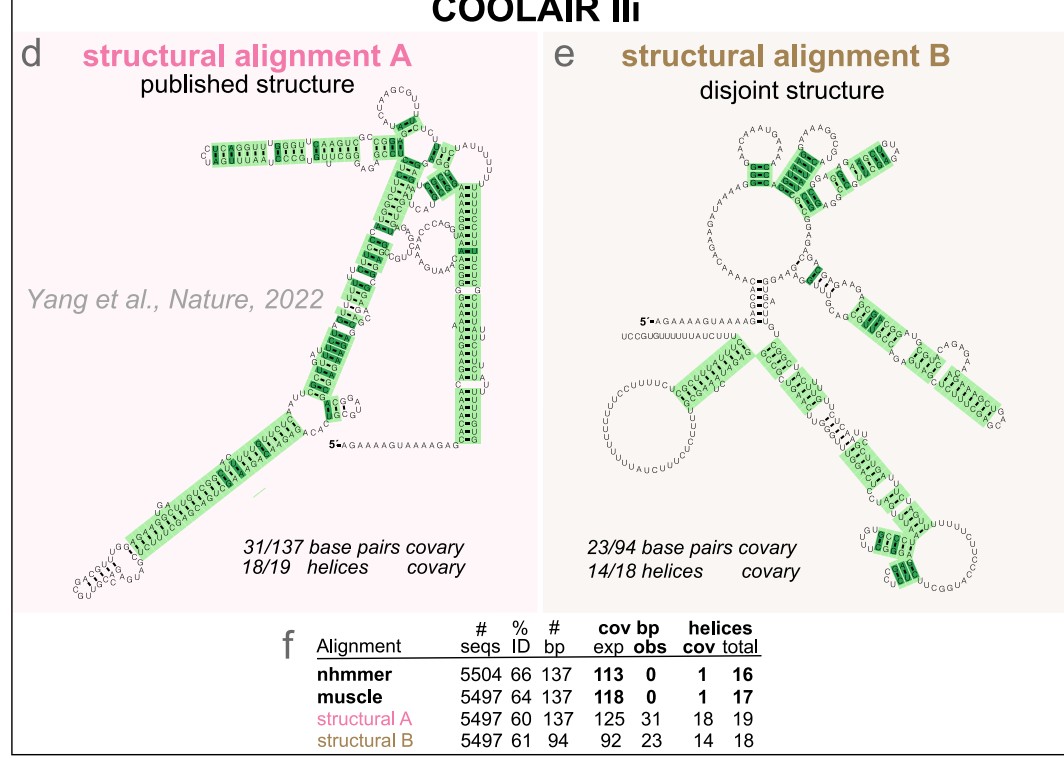

**d** structural alignment A
published structure

*Yang et al., Nature, 2022*

31/137 base pairs covary
18/19 helices covary

**e** structural alignment B
disjoint structure

23/94 base pairs covary
14/18 helices covary

**f**

| Alignment | # seqs | % ID | # bp | cov bp exp | cov bp obs | helices cov | helices total |
|---|---|---|---|---|---|---|---|
| **nhmmer** | 5504 | 66 | 137 | **113** | **0** | 1 | 16 |
| **muscle** | 5497 | 64 | 137 | **118** | **0** | 1 | 17 |
| structural A | 5497 | 60 | 137 | 125 | 31 | 18 | 19 |
| structural B | 5497 | 61 | 94 | 92 | 23 | 14 | 18 |

**Fig 4. Tests for significant covariation need to not use covariation in producing their alignments.** Using NEAT1 and COOLAIR as examples of lncRNAs with proposed structures, I show two structural alignments for each, one for the published proposed structures ((**a**) [12] and (**d**) [27]) and another for alternative made up structures of comparable complexity but without any shared base pairs ((**b**) and (**e**)). The alternative structures were created by folding the two halves of reference sequences (the human sequence for NEAT1 and the *A. thaliana* sequence for COOLAIR) independently using the program

RNAfold (ViennaRNA [15]). The structural alignments were created using the program cmalign (Infernal [16]) after building a profile SCFG from the reference sequence/structure. Covariation analysis of the alignments reports significant helices in light green and significant base pairs in dark green. Covariation analysis is reported on the reference sequences. Comparison to non-structural alignments created with the methods nhmmer [17] and muscle [18] are given in (c) and (f).

published ones. Results in Fig 4d and 4e show how both structures equally induce spurious covariation, and the majority of the helices seem to significantly covary. On the other hand, non-structural alignments created by nhmmer and muscle mostly remove the signal (Fig 4f).

*HOTAIR, MEG3, XIST and ncSRA lncRNAs show little evidence of a conserved structure.* The proposed structures for lncRNAs HOTAIR [20], MEG3 [21], XIST [22–24] and ncSRA [26], obtained by a combination of chemical probing with a RNA structure prediction algorithm, report hundreds of base pairs arranged in scores of putative helices. A statistical analysis of the associated alignments indicates that the claims of evolutionary structural conservation for these lncRNA structures were not statistically significant [1]. In fact, the alignments show evidence against a conserved RNA structure [2]. Helix-level covariation corroborates those findings (Fig 3b). The alignments of HOTAIR, MEG3, XIST and ncSRA all have enough power to expect to find at least 14% of the base pairs to significantly covary, but only one or zero covarying base pairs are observed per structure.

Helix aggregation analysis in Fig 3b and Fig 3 Insert shows a few helices with no significantly covarying base pairs that achieve aggregated E-values smaller than 0.05. In particular, three out of the 98 proposed helices in HOTAIR are marginally significant (Lancaster E-values: 0.01, 0.01, 0.02), and 2/62 helices for the RepA region of XIST are also significant (Lancaster E-values: $2.9 \times 10^{-5}$, 0.003). Those few helices do not provide support for the full structures, but they suggest a direction to follow as to producing alignments that could identify additional evolutionary signal.

*Cyrano and LincRNAp21 alignments lack power to make any evolutionary inference.* Cyrano is a long intergenic non-coding RNA (lincRNA) first identified in zebrafish with homology in vertebrates [35]. An RNA structure said to be evolutionarily conserved from fish to mammals has been proposed for Cyrano [19]. Using the UCSD 100 vertebrate genome database [36], I have constructed an alignment of Cyrano homologs using nhmmer. The alignment is very conserved with 91% average pairwise identity and it has no covariation or power for any of the 38 base pairs, or any of the 6 helices.

LincRNAp21 is a vertebrate lncRNA that includes inverted repeat *Alu* elements. Two structures have been proposed for the sense and antisense lincRNAp21, and an alignment of eight primate sequences has been produced [25]. The helix-level covariation analysis of these alignments revels that these alignments lack the power to make any evolutionary inference about a conserved RNA structure.

## Discussion

I have introduced a robust measure of RNA structural covariation that extrapolates from base-pair covariation to helix-level covariation. Helix-level covariation inference is the result of aggregating the p-values obtained for individual base pairs. Base-pair p-values are calculated using R-scape, which simulates the null distribution of covariation due to the phylogeny alone without any structural constraints. I observe than the Lancaster method that uses weights with information about covariation power is the most accurate aggregation method.

Using real and synthetic RNA alignments, I show that helix aggregated E-values are more accurate than base-pair E-values. In particular, I compare the sensitivity/specificity of using

helix E-values to assign significance to a helix to an *ad hoc* no-aggregation method that selects helices with a specified minimum number of significantly covarying base pairs. The benchmarks show that helix-aggregation reports better sensitivity for the same number of false positives.

Using helix-level covariation, I have re-examined the evidence for a conserved RNA structure present in several lncRNAs with proposed structures. For some lncRNAs, significance is observed for a small fraction of the helices in the absence of base pair significant covariation. In most cases, these constitute a very small fraction of all proposed helices. While this analysis does not provide support for the complete and usually large proposed structures, it provides a path to further explore a structural function for the RNA.

Using helix-level covariation, I have characterized a statistical pitfall in the examination of evidence for a conserved RNA structure. A test for significant covariation need to not use covariation in producing their alignments. Aligning to a covariance model (profile SCFG) assumes that the covariances in the alignment are true. Taking a CM-generated alignment and then testing it for whether its covariances are significant is invalid circular logic. This circular argument is more noticeable for helix-level covariation and lncRNAs where substantial amount the sequence heterogeneity may allow for radically different alignments. Using the lncRNAs COOLAIR and NEAT1, I have shown how by constructing structural alignments, it is possible to force covariation for arbitrary sets of proposed base pairs. Accurate evolutionary inference for long RNA transcripts with variable degrees of homology requires careful attention to these issues.

Helix-level covariation is a robust measure of covariation in RNA helices, but it cannot alone replace base-pair level covariation. Base-pair level covariation is still fundamental to capture covariation from structural elements that do not arrange into helices such as non Watson-Crick base pairs, triplets, and other 3D interactions. Moreover, reporting helix-level covariation–unlike base-pair level covariation–requires that a structure is proposed. As I have shown here, that can be a source of artifacts when the structure is just a prediction. Both helix and base-pair level covariation should be considered in combination, and consequently R-scape reports them both superimposed in light and dark green respectively. Next, steps will be taken to improve RNA structure prediction by statistically significant base-pair covariation incorporating helix-level covariation [37].

## Materials and methods

### Methods to aggregate p-values

I aim to aggregate the p-values of all base pairs in a given RNA helix. A helix is functionally defined as a collection of contiguous pairs, where I allow a maximum of two unpaired residues within the helix (that is, either a $1 \times 1$ internal loop that could represent a non Watson-Crick base pair or a 1 or 2 nucleotide bulge). The p-values of the individual base pairs in an alignment are themselves calculated from the null distribution of phylogenetic only covariation obtained using the software R-scape [1].

Given p-values $(p_1, \ldots, p_N)$ from $N$ independent null hypothesis tests, the p-values are uniformly distributed,

$$p_n \sim \text{Uniform}[0, 1].$$

The fact that p-values following the null hypothesis are uniformly distributed is part of a more general argument. For any hypothesis H, described by a continuous probability

distribution $P_H$ on $X$,

$$H : x \sim P_H(x), \quad x \in X,$$

with cumulative distribution function (CDF) $F_H(x) =: P_H(X \leq x)$, the random variable $F_H(X)$ defined on $r \in [0, 1]$ follows the uniform distribution Un $[0, 1]$.

I show that by observing that the CDF on $F_H(X)$ is given by

$$P(F_H(X) \leq r) = P(X \leq F_H^{-1}(r)) = F_H(F_H^{-1}(r)) = r,$$

thus $F_H(X)$ is Un $[0, 1]$, and $p_n = 1 - F_H(p_n)$ is also Un $[0, 1]$.

I considered four statistics to combine base pair p-values. Those have been selected because of their simplicity and mathematical convenience as they are all based on statistics that have closed-form distributions from which I can easily calculate an aggregated p-value. Those four p-value aggregation methods are:

***Fisher method*** [5]

The Fisher method uses the statistic,

$$T_f(p_1 \ldots p_N) = \sum_{n=1}^{N} -2 \log(p_n) \geq 0.$$

Under the assumption of independence for the $N$ tests, $T_f(p_1 \ldots p_N)$ follows a chi-squared distribution with $2N$ degrees of freedom, $\chi^2_{2N}$.

That result follows because the negative log of a Uniform[0,1] distributed variable follows an exponential distribution, that after scaling by $-2$ follows a $\chi^2_{df=2}$ distribution with 2 degrees of freedom, and the sum of $N$ independent $\chi^2_{df=2}$ values follows a $\chi^2_{df=2N}$ distribution with $2N$ degrees of freedom.

Then, a Fisher combined p-value is calculated as

$$\text{p-value}_f(p_1 \ldots p_N) = P_{\chi^2_{2N}}(T_f > T_f(p_1 \ldots p_N)) = 1 - F_{\chi^2_{2N}}(T_f(p_1 \ldots p_N)),$$

where $F_{\chi^2_{2N}}$ is the cumulative distribution (CDF) of the chi-squared distribution with $df = 2N$ degrees of freedom.

***Lancaster method*** [6]

This method generalizes the Fisher method. Each p-value $p_n$ comes with an associated integer weight $w_n > 0$, such that the Lancaster statistic is defined as,

$$T_l(p_1 w_1 \ldots p_N w_N) = \sum_n F_{\chi^2_{w_n}}^{-1}(p_n),$$

where $F_{\chi^2_{w_n}}^{-1}$ is the inverse $\chi^2$ CDF with $df = w_n$.

Under the assumption of independence, $T_l$ follows a $\chi^2$ distribution with $df = \sum_n w_n$ degrees of freedom.

$$T_l \sim \chi^2_{\sum w}.$$

Then, a Lancaster combined p-value is calculated as

$$\text{p-value}_l(p_1 w_1 \ldots p_N w_N) = 1 - F_{\chi^2_{\sum w}}(T_l),$$

where $F_{\chi^2_{\sum w}}$ is the $\chi^2$ CDF with degrees of freedom $df = \sum_n w_n$.

For the particular case in which all weights are equal and have value $w_n = 2$, $\forall n$, the Lancaster method becomes the Fisher method.

To aggregate the p-values of base pairs in a RNA helix, I have used the Lancaster method weighted by the number of substitutions per base pair.

$$w_i = \text{substitutions}(i),$$

where the substitutions for a given base pair $i$ are calculated by R-scape using the inferred phylogenetic tree and the Fitch algorithm [38].

***weighted-Fisher method*** [8]

This method further generalizes Lancaster's by using arbitrary non-integer weights. It relies on the fact that the chi-square distribution with $df$ degrees of freedom is a particular case of the Gamma distribution

$$\Gamma(\alpha, \beta, x) = \frac{\beta^\alpha}{\Gamma(\alpha)} x^{\alpha-1} e^{-\beta x},$$

for the particular values $\alpha = df/2$ and $\beta = 1/2$.

That is,

$$\chi^2_{df}(x) = \Gamma(\alpha = df/2, \beta = 1/2, x).$$

The weighted-Fisher method then uses the statistic

$$T_{wf}(p_1 w_1 \ldots p_N w_N) = \sum_n F^{-1}_{\Gamma(w_n/2, 1/2)}(p_n),$$

with arbitrary real weights $w_n \geq 0$.

Under the assumption of independence,

$$T_{wf} \sim \Gamma\left(\sum_n w_n/2, 1/2\right).$$

Then, a weighted-Fisher combined p-value is calculated as

$$\text{p-value}_{wf}(p_1 w_1 \ldots p_N w_N) = 1 - F_{\Gamma(\Sigma w/2, 1/2)}(T_{wf}).$$

To aggregate the p-values of base pairs in a RNA helix, I have used the weighted-Fisher method weighted by the power of each base pair. The weights are calculated as

$$w_n = 2 \frac{\text{power}_n}{\langle \text{power} \rangle}, \quad \text{where} \quad \langle \text{power} \rangle = \frac{1}{N} \sum_{n=1}^{N} \text{power}_n,$$

and the power of covariation per base pair, $\text{power}_n$, is calculated by R-scape.

As in the unweighted Fisher method (which corresponds to constant weights $w_n = 2$), these Fisher weights are normalized to a constant weight $\sum_n w_n = 2N$. In contrast, the Lancaster method assigns integer weight that result in much larger aggregated dimensions.

***Šidák method*** [7]

The Šidák test depends only on the minimum p-value $p_{min} = min(p_1, \ldots, p_N)$ and the number of tests $N$. It calculates the probability that under the null hypothesis, the minimum p-value is less or equal than $p_{min}$. Introducing $P_n$ as the probability distribution of p-values for

test $n$,

$$\text{p-value}_s(p_{min}, N) = P\left(\bigcup_{n \in N} (P_n \leq p_{min})\right) = 1 - P\left(\bigcap_{n \in N} (P_n > p_{min})\right)$$

Under the assumption of independence which results in the p-values being uniformly distributed, $P(P_n > p_{min}) = 1 - p_{min}$, I have

$$\text{p-value}_s(p_{min}, N) = 1 - \prod_n P(P_n > p_{min}) = 1 - (1 - p_{min})^N.$$

For small values of $p_{min}$, I can use the Taylor expansion approximation

$$\text{p-value}_s(p_{min}, N) = 1 - (1 - p_{min})^N \sim N p_{min},$$

which is equivalent to a Bonferroni correction.

## Construction of the sensitivity/specificity benchmarks

To obtain the Rfam-derived set of 326 structural RNAs used in the benchmark, I applied the following selection criteria: (1) remove all 1,530 micro-RNA families and all 1,938 snoRNA families, due to their relatively low structural complexity; (2) remove families with fewer than 40 sequences in the seed alignment; and (3) remove families where the average percentage identity in the alignment is larger than 95%. Using these criteria, I selected 326 out of 4,108 structural RNA families in Rfam 14.9. A list of all 326 families is in S1 Data.

For each of the 326 real alignments selected from Rfam, I generated two sets of synthetic alignments under two different evolutionary models: one set of 19,079 alignments preserving the phylogeny and structural constraints, and another set of 19,068 alignments preserving the phylogeny but not the structural constraints. These two sets of synthetic alignments constitute the "positive" and "decoy" sets of the synthetic benchmark. For the real benchmark, I also generated another set of 12,680 of decoy alignments produced by R-scape from the 326 trusted multiple sequence alignments to execute the null hypothesis of similar evolutionary histories in the absence of any specific covariation between the positions.

In order to generate realistic decoy structures, the consensus structure of a real RNA was transferred to the inferred ancestral sequences at the origin of each decoy alignment. While that structure is not used in the evolutionary process that generates the synthetic decoys nor in the generation of the R-scape null alignments, the location of the base pairs is remembered throughout the process. That results in decoy alignments with an annotated decoy structure that reflects similar distributions of number of base pairs, number of helices and number of base pair per helix than the original alignments.

Details of the characteristics of all the alignments and their structures (real or synthetic, positive or decoys) are presented in S1 Fig. S1 Fig (top) provides statistics for the alignments. The pairwise percentage identity ranges from 45%-90% with a median of 60–70% for all four types of alignments. S1 Fig (bottom) provides statistics on the alignment consensus structures. I observe that the R-scape null alignments are overall much shorter than their real RNAs counterparts. That is because R-scape disregards alignment columns with more than 70% gaps by default. Nevertheless, notice that the remaining columns preserve the structure, and the R-scape null alignments have similar number of base pairs and helices than the other types of alignments.

### Aggregated helix p-values in null alignments are uniformly distributed

I tested the null hypothesis as implemented in the benchmarks. In S5(a) Fig, I observe that the distribution of helix p-values for the benchmark decoys are uniformly distributed as it should be expected under the null hypothesis. For completeness, S5(b) Fig shows the distribution of base pair p-values for the benchmark decoys which are also uniformly distributed.

I also tested that under the null hypothesis the p-values of base pairs in the same helix are uncorrelated. For 1000 helices selected at random from the collection of decoy alignments, S5(c) Fig displays a scatter plot of the p-values for two base pairs taken at random from each of the helices. The same-helix base-pair p-values appear independent by visualization.

### Alignments

The 362 structural RNA alignments used in the real RNAs benchmark and in Fig 3 are Rfam seed alignments [14]. The list of the 362 Rfam families is provided in S1 Data.

The provenance of the alignments and structures for the 8 lncRNAs is as follows. The COOLAIR proposed structure is provided in [10, 27] (Fig 3a), and several alignments were produced for this work. I searched the NCBI database of plant genomes for homologs of the *Arabidopsis thaliana* COOLAIR class II.i isoform sequence using nhmmer [28] with an E-value cutoff of $10^{-5}$. The nhmmer alignment consists of the alignment of all the significant hits. The other non-structural alignment was produced realigning the nhmmer homologous sequences with the program MUSCLE [18] using default parameters. For simplicity, I searched only with a contiguous genomic section (positions 137–550) of the COOLAIR class II.i isoform (656 nucleotides) antisense to the first exon of the FLC gene and that contains the largest structural element in the proposed structure (Fig 2a [17]).

The two COOLAIR structural alignments were created by aligning all homologous sequences using the program `cmalign` from the package Infernal [16]. In one case, I used the *A. thaliana* COOLAIR class II.i isoform sequence annotated with the structure proposed in previous studies [27], in the other case, I used the same sequence annotated with a structure created using the program `RNAfold` from the ViennaRNA package [15] after splitting the sequence in two halves folded independently. This results in a proposed structure with 92 base pairs all different from the 137 base pairs of the [27] structure.

The Cyrano RNA structure was obtained from [19], and a Cyrano alignment using nhmmer was produced for this work. The HOTAIR alignments annotated with the proposed structures were provided by the authors [20]. We also generated structural HOTAIR alignments using Infernal's `cmalign`, for an Infernal covariance model created with the human/ sequence structure from [20]. These structural alignments do not provided any significant change in covariation relative to that observed in the input alignments.

The proposed structure and alignments for MEG3 were obtained from [21]. NEAT1 structural alignments were provided by the authors who reported the structures [12]. NEAT1 non-structural alignments were produced for this work by aligning all sequences to a one-sequence nhmmer model created for the human NEAT1 sequences.

Several different XIST alignments were tested, some of them covering the RepA region [22–24], and others covering 10 different homology regions covering the whole XIST RNA [2]. The lincRNAp21 alignments were provided by the authors who reported the structures [25]. The ncSRA proposed structure was obtained from [26], and the alignment was created in [1].

All alignments are provided in S1 Data.

### Software implementation and availability

The aggregated p-value method has been implemented in the software program R-scape version 2.0.0.p (December, 2022). Using any of the options to report the helix-aggregation methods, R-scape produces a file `.helixcov` with all the information about the helix-level analysis. Details are given in the R-scape userguide provided in the supplemental material. The code is available from the supplemental materials of this manuscript, from the R-scape website eddylab.org/R-scape/, and from the lab website rivaslab.org. Data used in this manuscript, including alignments, are part of the supplemental material (referred to as S1 Data in the manuscript) http://rivaslab.org/publications/Rivas23a/supplemental_material.tar.gz.

### Software and database versions

I used the Rfam database version 14.9 (November, 2022). I used the NCBI collection of 144 plant whole genomes downloaded on August 20, 2022. I used the alignment method MUSCLE v3.8.31 [18], and the program RNAfold v2.4.14 from the ViennaRNA package [15].

### Supporting information

**S1 Fig. Statistics of the RNA alignments and structures used in the benchmarks.** Statistics are provided for the positive alignments/structures in the real and synthetic benchmarks as well as for the decoy alignments/structures.
(EPS)

**S2 Fig. Benchmark of helix-covariation using false discovery rate (FDR).**
(EPS)

**S3 Fig. Benchmark of helix-level covariation sensitivity and specificity using R-scape's one-set test. (a, b)** Sensitivity and specificity benchmarks. **(c, d)** Helix p-value accuracy.
(EPS)

**S4 Fig. Benchmark of base pair covariation.**
(EPS)

**S5 Fig. Aggregated helix p-value statistics under the null hypothesis of no structural covariation.** (**a**) Helix aggregated p-values follow a uniform distribution under the null hypothesis. (**b**) Base pair p-values follow a uniform distribution under the null hypothesis. (**c**) Independence of the base pair p-values in a helix under the null hypothesis. I selected 1000 random null helices per benchmark, and for each helix I plot the p-values of two randomly selected base pairs in the helix.
(EPS)

### Acknowledgments

Thanks to Sean R. Eddy and Eric P. Nawrocki for a critical reading of the manuscript.

### Author Contributions

**Conceptualization:** Elena Rivas.

**Data curation:** Elena Rivas.

**Formal analysis:** Elena Rivas.

**Funding acquisition:** Elena Rivas.

**Investigation:** Elena Rivas.

**Methodology:** Elena Rivas.

**Project administration:** Elena Rivas.

**Resources:** Elena Rivas.

**Software:** Elena Rivas.

**Validation:** Elena Rivas.

**Visualization:** Elena Rivas.

**Writing – original draft:** Elena Rivas.

**Writing – review & editing:** Elena Rivas.

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
