## [Decision Letter · Decision Letter 0]

30 May 2023

Dear Dr Rivas,

Thank you very much for submitting your manuscript "RNA covariation at helix-level resolution for the identification of evolutionarily conserved RNA structure" for consideration at PLOS Computational Biology. As with all papers reviewed by the journal, your manuscript was reviewed by members of the editorial board and by several independent reviewers. The reviewers appreciated the attention to an important topic. Based on the reviews, we are likely to accept this manuscript for publication, providing that you modify the manuscript according to the review recommendations.

All three reviewers were extremely positive about your work. I must say that I rarely see such positive unanimity among reviewers, and I congratulate you on excellent research and writing. I am deciding for Minor revision to provide you with the opportunity to take into accoun the comments of the reviewers. None of these are compulsory, but I feel that several would improve your manuscript, and I hope that you will feel the same.

Sincerely,

Marc Robinson-Rechavi

Academic Editor

PLOS Computational Biology

Arne Elofsson

Section Editor

PLOS Computational Biology

Reviewer's Responses to Questions

**Comments to the Authors:**

Reviewer #1: In this article, the author introduces an update to the R-Scape software package. Previous versions of R-Scape utilized a statistical test to determine the covariance of base paired RNA structures in the presence of variation (e.g., random mutation). This most recent update expands upon this idea, aggregating base pairs into helix-level structures and providing a measurement of their covariance. Overall, this was an excellent manuscript that presents a very useful new approach for evaluating the evolutionary impact of RNA structure on sequence and for aiding our understanding of RNA structure/function. The author performed extensive testing of the new version of R-scape including and in-depth analysis of lncRNA structure, which provides further evidence in support of her previous findings regarding the significance of proposed covariation in lncRNAs. I think this is important work as it pushes for more rigor and care in the use of evolutionary arguments for RNA structure/function and will be of wide interest to the readership of PLoS Computational Biology. I have only a few comments below.

The author argues provides evidence that the new helix aggregated metrics are more robust than the base-pair metrics. It would be helpful if they could provide suggestions or guidance to readers on what the roles of base pair vs helix level metrics should be: i.e., should each metric be calculated and assessed or are helix level metrics to replace base pair level ones? Are there cases where base pair level metrics could be better?

When the author writes, “I observe that all methods except for Šidák report higher sensitivity for the same number of false positives than taking as supported all helices with at least one covarying base pair.” do they mean that the number of false positives doesn't change with different values of k but sensitivity does, or that sensitivity divided by false positives grows?

Minor:

The manuscript is heavily written in first-person, maybe “depersonalize” it a little bit.

Figure 2 (a, b) has a space but (c,d) does not

The light blue font is a bit hard to read on Fig. 2.

A few noted minor grammar issues: e.g.

“The Lancaster weights allow (one) to incorporate additional”

“four aggregation[s] methods”

“Two structure[s] ha(ve) been proposed”

"explore a[n] structural function"

Reviewer #2: In Elena Rivas PhDs' research article, "RNA covariation at helix-level resolution for the identification of evolutionarily conserved RNA structure," she presents a novel methodology for recognizing RNA structures that have been evolutionarily conserved. The paper highlights a new technique for assessing statistically significant covariation at the RNA helix level, which is achieved by combining the significance and power of covariation calculated at the base pair level. Rivas begins by discussing R-scape, a tool developed to identify base pairs in RNA sequence alignments that covary beyond phylogenetic expectation. R-scape traditionally treats base pairs as independent entities, but Rivas highlights the fact that RNA base pairs do not exist in isolation—they stack together to form helices, which together create the complete structure of the RNA. It's these helix-forming Watson-Crick base pairs that are believed to carry most of the covariation signal in an RNA structure.

Introducing a new covariation measure that aggregates these signals at the helix level, Rivas compares its performance to the conventional base-pair level approach. She demonstrates that this new technique improves sensitivity in detecting evolutionarily conserved RNA structure without compromising specificity. The paper then discusses the concept of p-value aggregation, where a group of base pairs with borderline covariation evidence may collectively yield a more robust helix signal. This is achieved by aggregating the p-values of individual marginally significant pairs. Rivas asserts that this aggregation process doesn't decrease sensitivity; even a single base pair with a strong covariation signal can make the entire helix significant, irrespective of the significance of other base pairs in the helix.

Rivas also establishes a benchmark to evaluate the performance of the helix-level significant covariation test. This benchmark includes real structural RNAs, synthetic structural RNAs, and decoy RNA alignments that lack structure. The results of the R-scape covariation analysis are combined and ranked by helix E-value, and the ranked list of helices is used to plot the fraction of true positive helices detected against an increasing threshold of mean false positives per query alignments.

Finally, Rivas reappraises the evolutionary evidence for a subset of long non-coding RNAs (lncRNAs) using the new helix-level approach. This evaluation uncovers an artifact that arises when covariation is employed to build an alignment for a hypothetical structure, which is then tested for whether its covariation significantly supports the structure. This helix-level reassessment strengthens the argument against these lncRNAs having a conserved secondary structure.

All of these results are important and impactful steps forward for RNA researchers.

Minor Points:

Helix level conservation is a powerful way to build consensus structures (https://academic.oup.com/bioinformatics/article/22/24/2988/208807), does this metric aid R-scape in predicting new structures, either with cacofold or otherwise?

Now that R-scape is moving towards helix-level statistics, is there any chance covariation/consistency metrics (like the previously misused RAFS metric) can be reevaluated for their utility?

Didn’t proofread extensively but found minor typos, like “mayority” instead of “majority” on page 9 for example.

Reviewer #3: The author is looking to extend the work presented in Rscape on statistical identification of evolutionarily conserved RNA structure from the base pair level to the helix level, as RNA base pairs do not occur in isolation.

The author explores four statistical methods to aggregate p-values: Fisher, Lancaster, Sidak, and a weighted Fisher approach.

The p-value aggregation approach is hypothesized increase sensitivity, and the author proposes a benchmark to assess whether that is associated with a loss of specificity, using real and synthetic RNA datasets, respectively with positive examples and decoys.

The author observes that sensitivity/specificity is increased in structured RNAs, offering higher sensitivity with lower false positive rates. As an application, the author looks at potentially conserved structured in well-known long non-coding RNAs, uncovering more precise information than was previously available with state of the art tools.

This paper is well structured, with a well defined objective and hypothesis, transparent and logical methods, clearly presented results and relevant exploratory applications. In particular, the construction of the decoys for the sensitivity/specificity benchmarks is very elaborate and strongly contributes to making the results more convincing.

My only suggestion would maybe suggest discussing why three of the four methods appear to perform very similarly (and extremely well) whereas sidak seems to underperform a bit, as this result raises some curiosity in a non-expert eye. While this is not necessary for the results presented to be convincing, it may help the discussion.

**Have the authors made all data and (if applicable) computational code underlying the findings in their manuscript fully available?**

Reviewer #1: Yes

Reviewer #2: Yes

Reviewer #3: Yes

PLOS authors have the option to publish the peer review history of their article (what does this mean?). If published, this will include your full peer review and any attached files.

Reviewer #1: **Yes: **Walter Moss

Reviewer #2: No

Reviewer #3: No

Figure Files:

Data Requirements:

Reproducibility:

References:

---

## [Editor Report · Decision Letter 1]

12 Jun 2023

Dear Dr Rivas,

We are pleased to inform you that your manuscript 'RNA covariation at helix-level resolution for the identification of evolutionarily conserved RNA structure' has been provisionally accepted for publication in PLOS Computational Biology.

Best regards,

Marc Robinson-Rechavi

Academic Editor

PLOS Computational Biology

Arne Elofsson

Section Editor

PLOS Computational Biology

---

## [Editor Report · Acceptance letter]

4 Jul 2023

PCOMPBIOL-D-23-00725R1 

RNA covariation at helix-level resolution for the identification of evolutionarily conserved RNA structure

Dear Dr Rivas,

I am pleased to inform you that your manuscript has been formally accepted for publication in PLOS Computational Biology. Your manuscript is now with our production department and you will be notified of the publication date in due course.

With kind regards,

Lilla Horvath
